# Soluble St2 Induces Cardiac Fibroblast Activation and Collagen Synthesis via Neuropilin-1

**DOI:** 10.3390/cells9071667

**Published:** 2020-07-10

**Authors:** Lara Matilla, Vanessa Arrieta, Eva Jover, Amaia Garcia-Peña, Ernesto Martinez-Martinez, Rafael Sadaba, Virginia Alvarez, Adela Navarro, Amaya Fernandez-Celis, Alicia Gainza, Enrique Santamaria, Joaquín Fernandez-Irigoyen, Patrick Rossignol, Faiez Zannad, Natalia Lopez-Andres

**Affiliations:** 1Navarrabiomed, Complejo Hospitalario de Navarra (CHN), Universidad Pública de Navarra (UPNA), IdiSNA, 31008 Pamplona, Spain; lara.matilla.cuenca@navarra.es (L.M.); varriet8@hotmail.com (V.A.); ej14025@bristol.ac.uk (E.J.); amaiagpu@hotmail.com (A.G.-P.); emartinezm@med.ucm.es (E.M.-M.); jr.sadaba.sagredo@navarra.es (R.S.); virginia.alvarez.asiain@navarra.es (V.A.); adela.navarro.e@gmail.com (A.N.); amaya.fernandez.decelis@navarra.es (A.F.-C.); alicia.gainza.calleja@navarra.es (A.G.); 2Departamento de Fisiología, Facultad Medicina, Instituto de Investigación Sanitaria Gregorio Marañón (IiSGM), Universidad Complutense, 28040 Madrid, Spain; 3Proteored-ISCIII, Proteomics Unit, Navarrabiomed, Institute for Health Research, Universidad Pública de Navarra, IdiSNA, 31008 Pamplona, Spain; esantamma@navarra.es (E.S.); jokfer@gmail.com (J.F.-I.); 4INSERM, Centre d’Investigations Cliniques-Plurithématique 1433, UMR 1116, CHRU de Nancy, French-Clinical Research Infrastructure Network (F-CRIN) INI-CRCT (Cardiovascular and Renal Clinical Trialists), Université de Lorraine, 54035 Nancy, France; p.rossignol@chru-nancy.fr (P.R.); f.zannad@chru-nancy.fr (F.Z.)

**Keywords:** sST2, neuropilin-1, collagen, fibroblast activation, NF-κB

## Abstract

Circulating levels of soluble interleukin 1 receptor-like 1 (sST2) are increased in heart failure and associated with poor outcome, likely because of the activation of inflammation and fibrosis. We investigated the pathogenic role of sST2 as an inductor of cardiac fibroblasts activation and collagen synthesis. The effects of sST2 on human cardiac fibroblasts was assessed using proteomics and immunodetection approaches to evidence the upregulation of neuropilin-1 (NRP-1), a regulator of the profibrotic transforming growth factor (TGF)-β1. In parallel, sST2 increased fibroblast activation, collagen and fibrosis mediators. Pharmacological inhibition of nuclear factor-kappa B (NF-κB) restored NRP-1 levels and blocked profibrotic effects induced by sST2. In NRP-1 knockdown cells, sST2 failed to induce fibroblast activation and collagen synthesis. Exogenous NRP-1 enhanced cardiac fibroblast activation and collagen synthesis via NF-κB. In a pressure overload rat model, sST2 was elevated in association with cardiac fibrosis and was positively correlated with NRP-1 expression. Our study shows that sST2 induces human cardiac fibroblasts activation, as well as the synthesis of collagen and profibrotic molecules. These effects are mediated by NRP-1. The blockade of NF-κB restored NRP-1 expression, improving the profibrotic status induced by sST2. These results show a new pathogenic role for sST2 and its mediator, NRP-1, as cardiac fibroblast activators contributing to cardiac fibrosis.

## 1. Introduction

The interleukin 1 receptor-like 1 (IL1RL1) gene belongs to the IL-1 receptor family and encodes the protein ST2. The alternative splicing of IL1RL1 results in multiple transcript variants, including the membrane-bound ST2L receptor for IL-33 and a soluble inhibitory decoy receptor, sST2. Briefly, the IL-33/ST2L system is cardioprotective in both physiological and pathological conditions, promoting cell survival and blocking profibrotic intracellular signaling [1,2]. An imbalance of sST2 levels is one key contributor to cardiac fibrosis, which, finally, may lead to the development of heart failure (HF). Indeed, sST2 is produced by both cardiac fibroblasts and cardiomyocytes in response to injury or stress [3]. In cardiovascular cells, data concerning sST2 effects are scarce. Most of the studies have shown sST2 only as a decoy receptor that prevents IL-33/ST2L signaling. Thus, sST2 blocks the antihypertrophic effects of IL-33 in angiotensin II or phenylephrine-treated cardiomyocytes [3]. In endothelial cells, sST2 abolishes the increase in adhesion molecules and proinflammatory cytokines induced by IL-33 [4]. Noteworthy, sST2 can signal independently of IL-33 sequestration. In-line with this, in vascular smooth muscle cells, recombinant sST2 increases collagen, fibronectin, transforming growth factor (TGF)-β and connective tissue growth factor (CTGF) [5]. Moreover, sST2 promotes mitochondrial fusion in human cardiac fibroblasts (HCF), increasing oxidative stress and the secretion of inflammatory markers via NF-κB [6]. In clinical populations, higher levels of circulating sST2 are associated with increased myocardial fibrosis, adverse cardiac remodeling and worse cardiovascular outcomes [7]. However, the specific role of sST2 as a direct inductor of cardiac fibroblast activation and the synthesis of profibrotic molecules, to our knowledge, has never been investigated.

Cardiac fibroblasts are the major cells responsible for tissue fibrosis. Fibroblasts can be activated by a wide range of stimuli, TGF-β being one of the most-studied [8]. The conversion of quiescent fibroblasts into activated myofibroblasts promotes enhanced extracellular matrix production, a key event in cardiac fibrosis [9]. Furthermore, activated fibroblasts promote cardiomyocyte hypertrophy and dysfunction via the release of profibrotic factors, such as TGF-β1, generating a vicious circle [10]. However, the cell biology and molecular mechanisms of fibroblast activation remain poorly studied.

The aim of this study was to provide a mechanistic assessment of sST2 on human cardiac fibroblast activation into profibrotic myofibroblasts, using a proteomic approach, and to unravel the resulting in vitro effects in a pressure overload rat model. 

## 2. Methods

### 2.1. Cell Culture

HCF were obtained from Promocell (Heidelberg, Germany) and maintained in medium Fibroblasts Media 3 supplemented with 10% FBS, 1-ng/mL FGF2 and 5-μg/mL rh-insulin according to the manufacturer’s instructions. At least 4 different HCF batches, corresponding to different donors (biological replicates), were used between passages 4–6. Cells were stimulated with human recombinant sST2 (2 μg/mL, R&D Systems, Abingdon, UK) for 24 h or with human recombinant NRP-1 (10^−10^–10^−8^ M, R&D Systems, Abingdon, UK) for 5, 10, 15, 30 and 60 min or 24 h. The NF-κB inhibitor BAY-11-7082 (Santa Cruz Biotechnology, Heidelberg, Germany) was added at 10^−6^ M for 30 min before stimulating with sST2 or NRP-1. Each biological replicate was assayed at least in triplicate for intra-assay variability and reproducibility. Technical replicates were averaged to provide a single value per each biological replicate. 

### 2.2. Animal Model

Adult male Wistar rats (weight = 250–300 g) were obtained from Harlan Ibérica (Barcelona, Spain). Rats were randomly distributed into two different groups: control rats (control; *n* = 10) and rats subjected to pressure overload (PO) by placing a 0.58-mm (internal diameter) tantalum clip banding the ascending and supravalvular aorta, as previously described (*n* = 7) [11,12,13,14]. The occlusion of the aorta continued for six weeks. Then, the animals were finally euthanized [11]. Body weight was measured once a week. Systolic blood pressure was basally estimated as well after 3 weeks and at the end of the study by using tail-cuff plethysmographin unrestrained animals. The Animal Care and Use Committee of Universidad Complutense de Madrid approved all experimental procedures, according to the guidelines for ethical care of experimental animals of the European Community (approval ID: CEA-UCM 77/2012). 

### 2.3. Mass Spectrometry Based-Quantitative Proteomics

Untreated cardiac fibroblasts and sST2-stimulated cardiac fibroblasts were compared by a shotgun comparative proteomic analysis using iTRAQ (isobaric Tags for Relative and Absolute Quantitation). Global experiments were carried out with whole cell lysates from four biological replicates (*n* = 4) in each experimental condition. Peptide labeling, peptide fractionation and mass spectrometry analysis were performed as previously described [15,16]. After MS/MS analysis, protein identification and relative quantification were performed with the ProteinPilot™ software (version 4.5; AB Sciex Spain S.L., Madrid, Spain) using the Paragon™ algorithm as the search engine. Although relative quantification and statistical analysis were provided by the ProteinPilot software, an additional 1.3-fold change cutoff for all iTRAQ ratios (ratio < 0.77 or >1.3) and a *p*-value lower than 0.05 were selected to classify proteins as up- or downregulated (at least in two of three biological replicates). Proteins with iTRAQ ratios below the low range (0.77) were considered to be under-expressed, whereas those above the high range (1.3) were considered to be overexpressed. Proteomics validation was assayed in the same samples used for proteomics, and 5 additional biological replicates were included (*n* = 5). In addition, we assessed the expression of soluble NRP-1 in conditioned media randomly selected from different biological replicates. Each biological replicate was assayed at least in triplicate. Using a higher number of samples (at least 2 more HCF batches) was justified to avoid endowing statistical error type I, as no additional methods were used to assess sNRP-1. 

### 2.4. CRISPR/Cas9 Genome Editing Mediated Deletion of NRP-1

NRP-1 expression was knocked down by CRISPR/Cas9 (clustered regularly interspaced short palindrome repeats) guided genome editing. Cells were seeded into 6-well plates at 70% confluence and transfected with a pool of three plasmids, each encoding the Cas9 nuclease and a target-specific 20-nt guide RNA (gRNA) designed for maximum gene editing efficiency according to the manufacturer’s instructions (Santa Cruz Biotechnology, Heidelberg, Germany). Scramble gRNA CRISPR/Cas9 plasmid was used as a control.

Once NRP-1 knockout was generated, cells were treated with sST2 for 24 h. Then, cell extracts and supernatants were collected to evaluate fibroblast-to-myofibroblast transformations and the expression of fibrotic markers. 

### 2.5. Western Blot Analysis

Aliquots of 20 µg of total proteins were prepared from cells and electrophoresed on SDS polyacrylamide gels and transferred to Hybond-c Extra nitrocellulose membranes (Bio-Rad, Hercules, CA, USA). Membranes were incubated with primary antibodies for: NRP-1 (Santa Cruz, Heidelberg, Germany), alpha smooth muscle actin (α-SMA) (Sigma/Merck Life Sciences S.L.U., Madrid, Spain), vimentin (Santa Cruz, Heidelberg, Germany), collagen-3 (Santa Cruz, Heidelberg, Germany), connective tissue growth factor CTGF (Abcam, Cambridge, UK), ST2 (Abcam, Cambridge, UK), NF-κB and phospho (Cell Signaling Technology, Leiden, The Netherlands), p42/44 MAPK and phospho (Cell Signaling Technology, Leiden, The Netherlands), IRAK 1/4 (Cell Signaling Technology, Leiden, The Netherlands), p38 MAPK and phospho (Cell Signaling Technology, Leiden, The Netherlands), IL-33 (Santa Cruz, Heidelberg, Germany), MyD88 (Santa Cruz, Heidelberg, Germany), AP-1 (Santa Cruz, Heidelberg, Germany) and TRAF-6 (Santa Cruz, Heidelberg, Germany). After washing, detection was made through incubation with peroxidase-conjugated secondary antibody and developed using an electrochemiluminescence (ECL) kit (GE healthcare, Thermo Fisher Scientific, UK). After densitometry analyses, optical density values were expressed as arbitrary units. All Western blots were performed at least in triplicate for each experimental condition. Stain-free detection was used as a loading control for ulterior normalizations. 

## 3. ELISA

Soluble NRP-1 (sNRP-1), fibronectin, collagen type I, galectin-3 (Gal-3), TGF-β, matrix metalloproteinase-1 (MMP)-1, MMP-2, tissue inhibitor of metalloproteinase (TIMP)-1 and TIMP-2 levels were quantified in cell supernatants following the manufacturer’s instructions (R&D Systems, Abingdon, UK). The results were normalized to the control condition. Data were expressed as a fold change relative to the control conditions.

### 3.1. Real-Time Reverse Transcription PCR

Total RNA was extracted with Trizol Reagent (Qiagen, Hilden, Germany) and was reverse-transcribed into single-stranded cDNA using a random hexamers master mix from Bio-Rad. Quantitative PCR (qPCR) analysis was performed using SYBR green PCR technology (Bio-Rad, Hercules, CA, USA) (Appendix A). Relative quantification was achieved with MyiQ software. HPRT, GADPH and β-actin were used as housekeeping genes for normalization. The relative expression (DDCT) of each selected gene product was calculated using the efficiency corrected calculation model and are shown as fold changes of the mRNA expression. All qPCRs were performed at least in triplicate for each experimental condition.

### 3.2. Statistical Analyses

Data are presented as scatter dot plots where horizontal line denotes a median value, and vertical lines denote the interquartile range (IQR). Normality of distributions was verified by means of the Kolmogorov–Smirnov test. The unpaired Student’s *t*-test or the Mann Whitney U test were used to assess statistical differences between two experimental conditions as appropriate. Data were analyzed using a one-way analysis of variance, followed by Tukey’s post hoc test analysis to assess specific differences among groups or conditions using GraphPad Software Inc. (GraphPad, La Jolla, CA, USA). Pearson’s coefficients were calculated to determine correlations. The predetermined significance level was *p* < 0.05. 

## 4. Results

### 4.1. sST2 Promotes Adult HCF Activations into Profibrotic Myofiboblasts

sST2 treatment (2 μg/mL) increased the expression of the myofibroblast activation markers α-SMA (*p* = 0.0010) and vimentin (*p* = 0.0007) (Figure 1A), as well as fibronectin secretion (*p* = 0.0088) (Figure 1B), at 24 h. Moreover, treatment with sST2 enhanced collagen type I secretion (*p* = 0.0001) (Figure 1C) without modifying collagen type III levels (Figure 1D). Additional profibrotic markers that were assessed, including Gal-3, CTGF and TGF-β, were significantly upregulated (*p* = 0.0295, *p* = 0.0056 and *p* = 0.0015, respectively) (Figure 1E,F). However, secreted levels of MMP-1, MMP-2 or TIMP-1 showed no differences among the experimental conditions, whereas TIMP-2 was greater secreted in sST2-treated cells than the control counterparts (*p* = 0.0368) (Figure 1G). See original Western blot images in Appendix A.

### 4.2. sST2 Upregulates NRP-1 in Adult Human Cardiac Fibroblasts

A proteome-wide analysis of total cell extracts using isobaric tags (iTRAQ) coupled to 2D nano-liquid chromatography tandem mass spectrometry was performed in HCF treated with sST2 as previously reported [6]. NRP-1 was identified as an upregulated protein upon 24 h of stimulation (Appendix A). Complementary analyses validated the increase of NRP-1 protein levels induced by sST2 (*p* = 0.0079) (Figure 2A). Moreover, the stimulation with sST2 enhanced (*p* = 0.0125) the secretion of sNRP-1 (Figure 2B). See the original Western blot images in Appendix A.

### 4.3. NF-κB Mediates sST2-Induced NRP-1 Expression and the Activation of Adult HCFs into Profibrotic Myofibroblasts

We have previously described that sST2 promotes NF-κB phosphorylation after 30 and 60 min of stimulation in HCF [6]. The specific inhibitor of NF-κB, BAY 11-7082, was able to prevent the increase in sNRP-1 protein levels (*p* = 0.0022) (Figure 3A). Moreover, the pretreatment with the specific inhibitor of NF-κB blocked sST2 effects on the expression of myofibroblast markers α-SMA (*p* = 0.00331) and vimentin (*p* = 0.0311) (Figure 3B), while no effects were reported for fibronectin expression (data not shown). Interestingly, the blockade of the NF-κB pathway also abolished the sST2-induced secretion of collagen type 1 (*p* = 0.017), as well as the expression of the profibrotic markers CTGF (*p* = 0.0003) and TGF-β (*p* = 0.0311) (Figure 3C–E). See original Western blot images in Appendix A.

### 4.4. sST2-Dependent Activation of Adult HCF into Profibrotic Myofibroblasts is Partially Mediated by NRP-1

NRP-1 knockdown HCFs were generated using CRISPR/Cas9 technology. Expression of NRP-1 was successfully downregulated by up to 80% compared to scramble at the transcript and protein levels (Figure 3F,G), in NRP-1 knockdown HCFs. sST2 was not able to elicit a myofibroblastic response in adult HCFs knocked down for NRP-1, as shown in Figure 3H. Accordingly, NRP-1 knockdown HCFs expressed significantly lower levels of α-SMA (*p* = 0.0001) and vimentin (*p* = 0.0069), despite sST2 stimulation, when compared to HCFs. These expression levels were comparable to those seen in control HCFs (Figure 3H). Moreover, sST2 failed to enhance collagen synthesis (*p* = 0.0011) in NRP-1 knockdown cells (Figure 3I). Whereas the NRP-1 knockdown blocked sST2-induced Gal-3 secretion (*p* = 0.0053) (Figure 3J), the treatment with sST2 increased CTGF in both the scramble and NRP-1 downregulated HCF (Figure 3K). See original Western blot images in Appendix A.

### 4.5. Exogenous NRP-1 Increased the Expression of the Activation and Fibrosis Markers in Adult HCFs

In order to understand the contribution of sST2-induced sNRP-1 to the activation of profibrotic myofibroblast phenotypes, adult HCFs were treated with a soluble human recombinant NRP-1. Exogenous sNRP-1 was supplemented at different concentrations (10^−8^ M, 10^−9^ M and 10^−10^ M) during 24 h to evidence the lowest concentration inducing myofibroblast activation. Exogenous recombinant NRP-1 increased the expression of the myofibroblast activation markers α-SMA and vimentin in a dose-dependent manner, reaching the statistical significance only for 10^−8^ M (*p* = 0.0011 and *p* = 0.0288, respectively), with no effect on fibronectin secretion (Figure 4A,B). Thereby, the 10^−8^ M NRP-1 dose was subsequently used to explore the downstream signaling elicited by sNRP-1 on myofibroblast differentiation. Moreover, the treatment with NRP-1 enhanced collagen type I secretion also in a dose-dependent manner (*p* = 0.0379 for 10^−8^ M) (Figure 4C). Regarding other profibrotic markers, Gal-3 expression was increased with a dose-dependent pattern (*p* = 0.0006 for 10^−8^ M) by sNRP-1 treatment (Figure 4D), whereas CTGF expression was not modified (Figure 4E). NRP-1-treated cells secreted similar levels of MMP-1, MMP-2, TIMP-1 or TIMP-2 as control cells (Figure 4F). See original Western blot images in Appendix A.

In order to investigate the intracellular pathways early activated by exogenous NRP-1 in HCF, cells were treated for 5, 10, 15, 30 and 60 min with 10^−8^ M NRP-1. Stimulation with NRP-1 induced NF-κB phosphorylation after 60 min of stimulation (*p* = 0.0420) (Figure 5A) without affecting the phosphorylation levels of p42/44 MAPK (Figure 5B), IRAK1/4 (Figure 5C) and p38 MAPK (Figure 5D). 

The specific inhibitor of NF-κB, BAY 11-7082, was able to blunt the effects of exogenously added NRP-1 on the expression of myofibroblast markers α-SMA (*p* = 0.0028) and vimentin (*p* = 0.028) (Figure 5E). Interestingly, the blockade of the NF-κB pathway also abolished the sNRP-1-induced secretion of collagen (*p* = 0.0028) (Figure 5F) and the expression of the profibrotic molecule Gal-3 (*p* = 0.0040) (Figure 5G). See original Western blot images in Appendix A.

### 4.6. Cardiac Expression of sST2 and NRP-1 is Induced in a Pressure Overload Model

The expression of sST2 and sNRP-1 has been measured in myocardial tissues from a PO animal model. Briefly, PO rats presented increased cardiac fibrosis characterized by enhanced levels of α-SMA (1.6-fold), fibronectin (2.0-fold), collagen type I (2.3-fold), TGF-β (1.9-fold) and CTGF (1.7-fold) protein expressions (for further information, please refer to citation [11]). Perivascular myocardial fibrosis was also enhanced in PO rats. Moreover, α-SMA, vimentin and fibronectin immunostainings were increased in PO rats, as compared to the controls (Appendix A). Interestingly, rats subjected to PO exhibited a greater expression of total ST2 and NRP-1 ((*p* = 0.0027 and *p* = 0.0068, respectively) (Figure 6A,B). A specific transcript analysis suggests that ST2 elevation results from sST2 over-expression (Figure 6C), while ST2L levels were not significantly different (Figure 6D). Interestingly, ST2 was positively correlated with the cardiac expression of NRP-1 (r = 0.633, *p* = 0.02, Figure 6E). It is worth to note that the IL-33 expression was unaffected by the experimental condition both at the transcript (Figure 6D) and protein levels (Figure 6F). Intracellular pathways activated by the IL-33/ST2L pathway showed similar MyD88 levels and lower TRAF-6 and AP-1 expressions in PO rats, as compared to the controls (Figure 6G). See original Western blot images in Appendix A.

## 5. Discussion

The purpose of this study was to investigate the effect of sST2 on fibroblast activation, collagen secretion and profibrotic marker productions in HCFs. sST2 enhanced the profibrotic myofibroblast differentiation, as demonstrated by the enhanced expression of fibrotic mediators and collagen type I secretion. Using a proteomic approach, NRP-1 has been identified as an upregulated protein by sST2. Our study shows that sST2 profibrotic effects are partly mediated by NRP-1, which emerges as a new inductor of fibroblast activations and a profibrotic phenotype in HCFs. In addition, we identified NF-κB as a central signaling pathway in the profibrotic effects of sST2 via NRP-1, and that was further validated in vivo.

Most of the sST2 previously defined effects suggest its function as a decoy receptor, inhibiting IL-33 effects [3,17,18]. However, sST2 could exert other effects independently of the sequestration of IL-33 [19]. In-line with this evidence, we previously described for the first time the deleterious effects triggered by sST2 on HCFs, by increasing the production of reactive oxygen species and inflammatory molecules [6]. In the present study, we expand our previous findings to show that sST2 also increased the myofibroblast activation markers, as well as collagen and other profibrotic markers, such as Gal-3, CTGF or TGF-β. Fibrosis is a fundamental component of the adverse structural remodeling of the myocardium present in a failing heart [20]. In HF patients, the sST2 measurement provides a strong serologic overview of the cumulative myocardial fibrotic process [21]. Our data reinforce the role of sST2 as a biomarker and, also, emerges as a biotarget of the fibrotic process, describing the mechanisms by which elevated sST2 could contribute to myofibroblast activation and collagen accumulation.

NRP-1 is a 130-kDa transmembrane protein identified as a coreceptor for multiple growth factors, including TGF-β1, where it could enhance responses to both the active and latent forms of this cytokine [22]. It is of special interest that the alternative splicing of mRNA encoding NRP-1 gives rise to several sNRP-1 (soluble NRP-1) isoforms [23]. sST2 enhanced both the membrane-bound and sNRP-1 in HCFs. This result was reinforced in the myocardium of PO rats, where increased ST2 levels correlated positively with the expression of NRP-1. Moreover, NRP-1 partly mediated the profibrotic sST2 effects in HCFs. Our data is in-line with other observations showing that NRP-1 could play a regulatory role in TGF-β1-induced fibrosis [24]. Interestingly, exogenous sNRP-1 induced myofibroblast activation, as well as collagen synthesis and the expression of profibrotic mediators. Although this is, to our knowledge, the first time that NRP-1 has been studied in HCFs, it has been previously suggested that NRP-1 promotes liver cirrhosis progression and its aggravation [25]. Moreover, the over-expression of NRP-1 promotes the endothelial-to-mesenchymal transition (EndMT) and associated fibrosis [26]. A positive correlation between NRP-1 levels, EndMT markers and profibrotic gene expressions has been reported in pancreatic ductal adenocarcinoma tissues [26]. In addition, the endothelial-specific loss of NRP-1 downregulates and inactivates TGF-β1 signaling, culminating in reduced CTGF and collagen type I [26]. Other results indicate that the NRP-1 antibody reduces fibrogenesis markers and inhibits fibrosis in hepatic stellate cells [22,25]. Thus, NRP-1 emerges as an attractive target in the context of fibrotic diseases.

NF-κB directly regulates the expression of fibrosis-related genes, including fibronectin and MMPs [27]. Moreover, NF-κB phosphorylation leads to the transformation of cardiac fibroblasts into myofibroblasts [28]. NF-κB phosphorylation also mediates the angiotensin II-stimulated profibrotic process [29] and regulates the levels of several profibrotic cytokines, leukocyte adhesion molecules and inflammatory molecules [30]. NF-κB has been previously described to contribute to the sST2 proinflammatory effects [6]. Our results, showing that NF-κB mediated the profibrotic effects induced by sST2, are not in agreement with a previous report showing that sST2 blocked IL-6 production by suppressing NF-κB activation in monocytic cells [31]. However, in the present study, both sST2 and NRP-1, a potential mediator of sST2 effects, did promote NF-κB phosphorylation in HCFs. Therefore, NF-κB seems to be a key regulator in controlling myofibroblast activation and collagen secretion elicited by both sST2 and NRP-1.

## 6. Conclusions

In summary, the present study shows that sST2 could affect myofibroblast activation, leading to an increase in collagen synthesis and profibrotic molecules in HCFs. Moreover, NRP-1, a molecule upregulated by sST2, emerges as a new, interesting target in cardiac fibrosis. Proinflammatory [6] and profibrotic effects triggered by sST2 via NF-KB certainly highlight the key role of the latter on cardiac fibrosis. New studies focusing on the interactions between sST2 and NRP-1 in experimental models are warranted.

## Figures and Tables

**Figure 1 cells-09-01667-f001:**
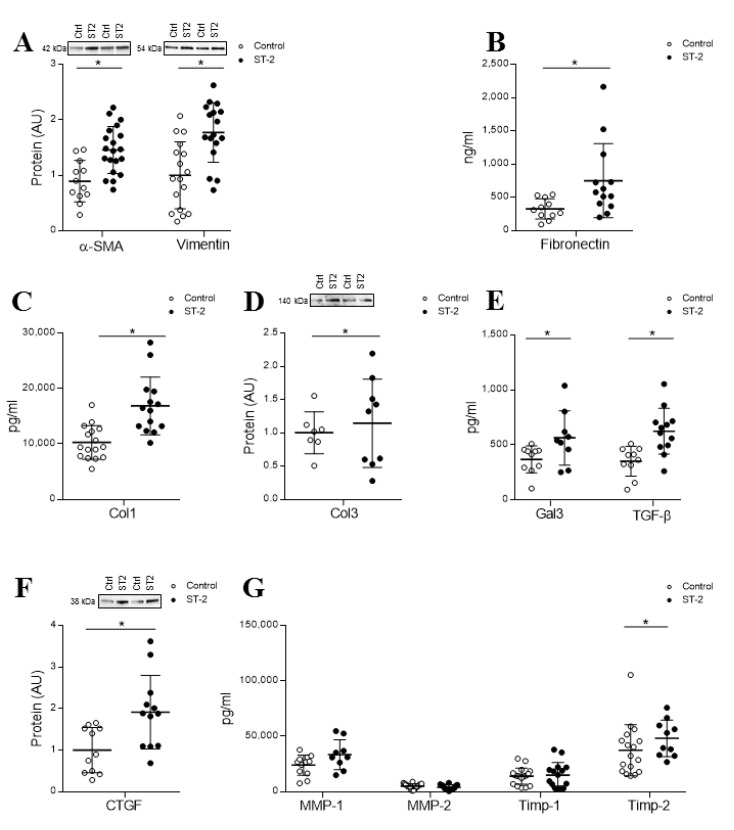
Soluble interleukin 1 receptor-like 1 (sST2) effects on fibroblast activation markers, collagens and profibrotic molecules in adult human cardiac fibroblasts (HCFs). The 24-h sST2 treatment (2 μg/mL) effect on α-SMA, vimentin (**A**), fibronectin levels (**B**), Col-1 (**C**) and Col-3 expression (**D**) in HCF. Effects of sST2 on the profibrotic markers Gal-3 and TGF-β1 (**E**) and CTGF (**F**). Effects of sST2 on MMP-1, MMP-2, TIMP-1 and TIMP-2 release (**G**). All conditions were performed at least in triplicate. Scatter dot plots represent the median and minimum to a maximum range of N = 7–20 replicates per condition. For Western blot experiments, protein densitometry was expressed in arbitrary units (AU) once normalized to stain-free for proteins. Representative blots have been displayed when appropriate. * *p* < 0.05 vs. control. α-SMA, smooth muscle actin; Col-1, collagen type I; Col-3, collagen type III; CTGF, connective tissue growth factor; TGF-β1, transforming growth factor beta; Gal-3, galectin-3; MMP, metalloproteinase and TIMP, tissue inhibitor of metalloproteinase.

**Figure 2 cells-09-01667-f002:**
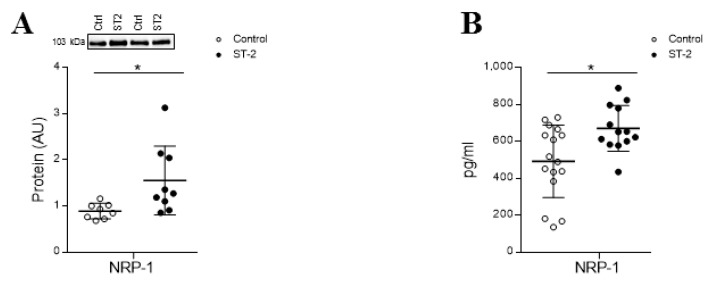
NRP-1 expression in HCFs treated with sST2. sST2 effect on intracellular NRP-1 expression (**A**) and sNRP-1 (**B**) in HCF. All conditions were performed at least in triplicate. Scatter dot plots represent the median and minimum to a maximum range of N = 8–16 replicates per condition. For Western blot experiments, protein densitometry was expressed in arbitrary units (AU) once normalized to stain-free for proteins. Representative blots have been displayed when appropriate. * *p* < 0.05 vs. control. NRP-1: neuropilin 1 and sNRP-1, soluble/secreted NRP-1 isoform.

**Figure 3 cells-09-01667-f003:**
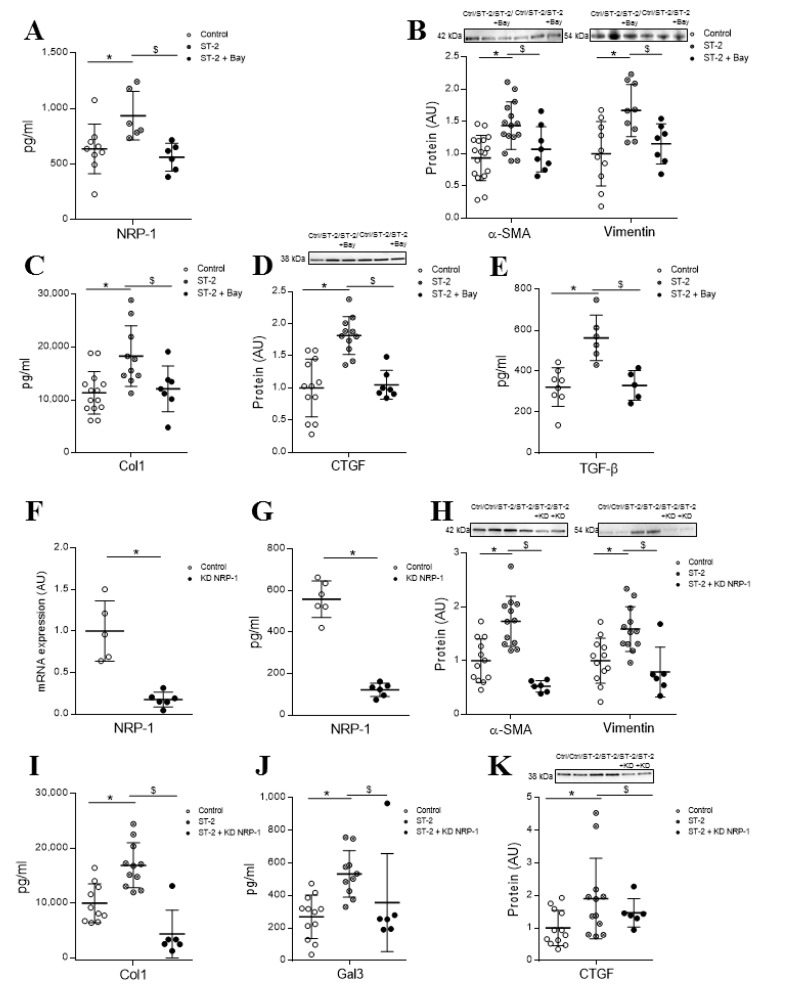
Intracellular pathways regulated by sST2 to promote its profibrotic effects. Nuclear factor-kappa B (NF-κB) mediates sST2 effects on NRP-1 expression (**A**), myofibroblast activation markers α-SMA and vimentin (**B**), profibrotic molecules Col-1 (**C**), CTGF (**D**) and TGF-β1 (**E**). Expression of NRP-1 at transcript (**F**) and protein (**G**) levels in NRP-1 knockdown HCFs. Induction of activation markers α-SMA and vimentin (**H**), Col-1 secretion (**I**) and the profibrotic molecules Gal-3 (**J**) and CTGF (**K**) by ST2 in NRP-1 knockdown HCFs. Scatter dot blots represent median and minimum to a maximum range of N = 5–18 replicates per condition. For Western blot experiments, protein densitometry was expressed in arbitrary units (AU) once normalized to stain-free for proteins. Relative mRNA expression is expressed in arbitrary units (AU) of the fold change after normalization to HPRT, β-actin and GADPH housekeeping genes. Representative blots have been displayed when appropriate. * *p* < 0.05 vs. control. ^$^
*p* < 0.05 vs. sST2. Statistical significance was studied by a one-way ANOVA analysis. NRP-1, neuropilin 1; α-SMA, alpha smooth muscle actin; Col-1, collagen type I; CTGF, connective tissue growth factor; TGF-β1, transforming growth factor beta and Gal-3, galectin-3.

**Figure 4 cells-09-01667-f004:**
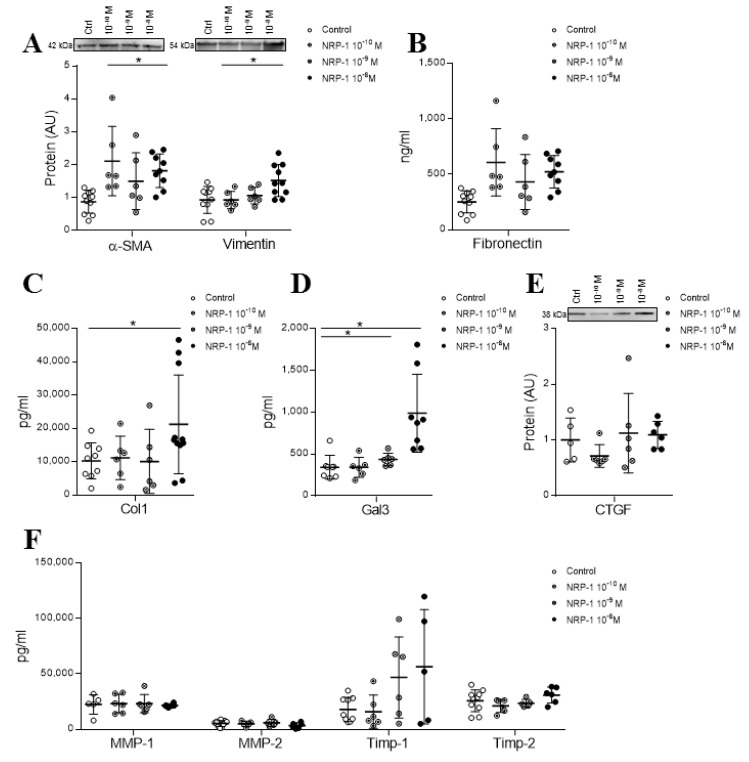
Effect of exogenous NRP-1 on fibroblast activation and profibrotic marker expressions in adult HCFs. Dose-dependent effects of exogenous NRP-1 were studied at a 24-h time point on myofibroblast activation markers α-SMA and vimentin (**A**); fibronectin (**B**); Col-1 secretion (**C**); profibrotic markers Gal-3 (**D**) and CTGF (**E**) and MMP-1, MMP-2, TIMP-1 and TIMP-2 (**F**). Scatter dot blot represents the median and minimum to a maximum range of N = 5–11 replicates per condition. For Western blot experiments, protein densitometry was expressed in arbitrary units (AU) once normalized to stain-free for proteins. Representative blots have been displayed when appropriate. * *p* < 0.05 vs. control group. α-SMA, alpha smooth muscle actin; Col-1, collagen type I; Gal-3, galectin-3; CTGF, connective tissue growth factor; MMP, metalloproteinase and TIMP, tissue inhibitor of metalloproteinase.

**Figure 5 cells-09-01667-f005:**
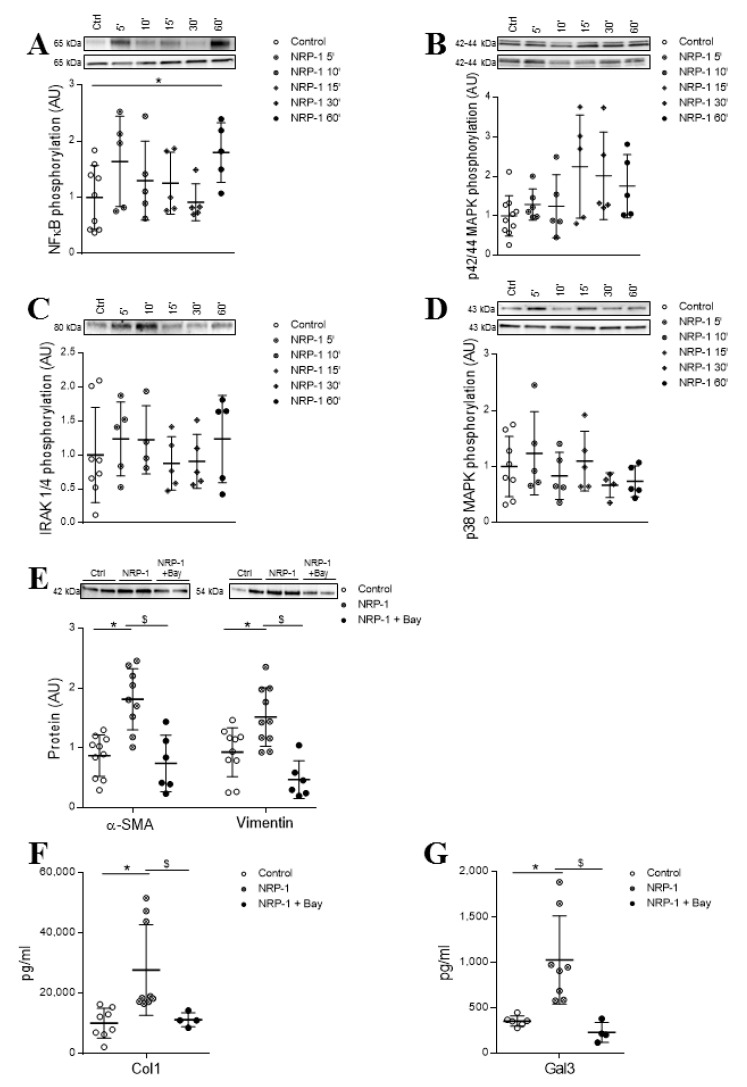
NRP-1 signaling pathways involved in its profibrotic effects in HCFs. Effect of 10^−8^ M NRP-1 on NF-κB (**A**), p42/44 MAPK (**B**), IRAK 1/4 (**C**) and p38 MAPK (**D**) phosphorylation. Effect of NRP-1 inhibition on fibroblast activation markers α-SMA and vimentin (**E**) and profibrotic markers Col-1 (**F**) and Gal-3 (**G**). Scatter dot blot represents the median and minimum to a maximum range of N = 4–0 replicates per condition. Protein phosphorylated upper and total protein expression lower. For Western blot experiments, protein densitometry was expressed in arbitrary units (AU) once normalized to stain-free for proteins. Representative blots have been displayed when appropriate. * *p* < 0.05 vs. control group. ^$^
*p* < 0.05 vs. NRP-1. α-SMA, alpha smooth muscle actin; Col-1, collagen type I and Gal-3, galectin 3.

**Figure 6 cells-09-01667-f006:**
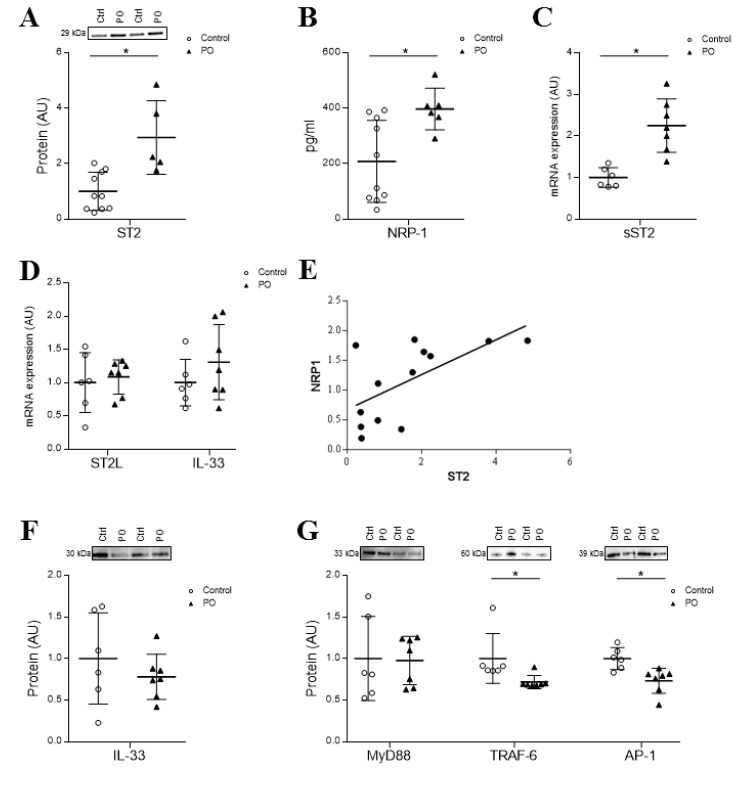
Cardiac expression of ST2 and NRP-1 in a pressure overload (PO) rodent model. Cardiac protein ST2 (**A**), NPR-1 protein expression (**B**) and sST2 mRNA (**C**) in myocardium samples of controls and PO rats. Gene expression of ST2L and IL-33 (**D**). Correlation between ST2 and NRP-1 expression (r = 0.633, *p* = 0.02) (**E**). Effects of PO on IL-33 protein (**F**) and MyD88, TRAF-6 and AP-1 protein expressions (**G**). Scatter dot blot represents the median and minimum to a maximum range of N = 5–10 animals per group. For Western blot experiments, protein densitometry was expressed in arbitrary units (AU) once normalized to stain-free for proteins. Relative mRNA expression was expressed in arbitrary units (AU) of fold change after normalization to HPRT, β-actin and GADPH housekeeping genes. Representative blots have been displayed when appropriate. * *p* < 0.05 vs. control group. ST2, interleukin 1 receptor-like 1; NRP-1, neuropilin-1; IL-33, interleukin 33; MyD88, Myeloid differentiation primary response 88; TRAF-6, tumor necrosis factor receptor associated factor 6 and AP-1, activator protein 1.

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
