# Peer review of "Soluble St2 Induces Cardiac Fibroblast Activation and Collagen Synthesis via Neuropilin-1"

_cells, 2020, doi:10.3390/cells9071667_

Round 1

Reviewer 1 Report

I thank the authors for the clarifications.

Author Response

Thank you

Reviewer 2 Report

This manuscript was very well written and the experimental design was well laid out.  Most of my concerns were regarding the quality of the western blots.

1.  For all western blots, the size of the blots should be expanded and multiple lanes loaded for each  protein being assayed- not a single band for each condition.

2. Loading controls should be shown to allow the reader to assess how equivalently the western blots were loaded.

3.  Immunocytochemistry should be performed to show organization of a-SMA in treated cells.

4.  For CRIPSR edited cells, their response to another pro-fibrotic agonist such as  TGFb should be assessed to see if it is specific response or a generalized response.

5. For NFKB,   subcellular fractionation, EMSA or immunofluorescent staining should be used to show activation and nuclear translocation of NFkB.

6. For the in vivo study, the western data is promising but histology staining to show degree of fibrosis and immunonhitochemistry to show cell specific and regional expression of markers is needed.

Author Response

We would like to thank the referee for his/her valuable comments and suggestions. We have done our best to address any technical and experimental concern and new experiments have been performed to make our results more consistent and sound. The revised manuscript has been accordingly modified where it was specifically requested or when further clarification was needed as raised by the referee’s concerns. If anything else needs to be additionally amended, we would be glad to do so after indication.

Round 2

Reviewer 2 Report

The quality of the western blots is concerning still.

Author Response

We appreciate all valuable comments and suggestions aiming to improve the quality of our original manuscript. We may apologise for any previous misunderstanding on the revision process. Technical and experimental concerns have been accordingly addressed to the best of our abilities. We hope to have properly answered your requests to grant its publication in your journal. In brief, the reviewed version of the manuscript includes additional supplementary material. Whole western blots of the representative images displayed in the main manuscript have been included together with their respective stain free images to provide more clear information regarding our methodology and experimental design. Note that we use pre-made SDS-PAGE with 18-26 lanes which are optimised to run as much experiments as possible. Whole blots included in the previous reviewed version have been cropped to only show those lanes corresponding to the experiments concerning this manuscript to avoid for further confusion. Representative bands of the whole statistical analysis have been replaced in the original manuscript, as appropriate, if new and improved blots have been obtained. The lanes have been labelled according to the experimental setting of the particular image provided. Moreover, we have indicated in the new Figure legends the number of dots per experimental condition resulting from such western blot. Whole images correspond to representative technical repeats of such western blot. We are aware that the quality of some blots might not be ideal but we have honestly shown our results. Statistics of densitometry band analysis reveals and supports both significant and non-significant data discussed in our manuscript. Nevertheless, when possible, we have tried to improve the quality of the blots in this new version of the manuscript to rise up our good faith and acknowledge the referee and editor’s comments. Besides, any modification in the manuscript has been tracked in ‘yellow’ for a more friendly evaluation. If anything else needs to be additionally amended, we would be glad to do so after indication.

Round 3

Reviewer 2 Report

I still have the same concerns regarding data quality.

This manuscript is a resubmission of an earlier submission. The following is a list of the peer review reports and author responses from that submission.

Round 1

Reviewer 1 Report

In the current study the authors investigated the effect of sST2 on human cardiac fibroblasts. Based on the already known effects of sST2 (fibrosis) they found (not surprisingly) that sST2 induces a differentiation phenotype of fibroblasts. Up to my knowledge the new finding of this study is the coupling to Nrp-1 via NFkB.

Major comments:

At least for me it would be essential to describe experiments in way that we can judge about the biological mechanism and at best repeat the experiments. In this direction the manuscript is not convincing.

Cells and cell culture: I spend a lot of time in the internet to try to figure out what kind of plating medium and growth and/or differentiation medium was used. The authors state that this is due to the manufacturer’s instruction but this is not convincing as the open instruction is not very detailed and gives you a couple of opportunities. Please precise the details. It may be critical to know whether this is a complete synthetic medium or whether it has any additions (such as serum). This may affect the results and also the conclusion as in vivo there is a competition between various factors.

Similarly, the animal model is not properly described. My understanding is that this is tissue that was left over from experiments in ref. 11. This is ok, but please state clearly that all further details (heart weight, Echo) are reported there. Furthermore, give at least information about duration of pressure overload, extent of overload, whether the hearts are already functionally impaired or not. This is for the conclusion essential. It would be more appropriate to show that by inhibition of sST2/Nrp-1 the disease prgress can be affected. Unless this, the experiment simply shows a similar regulation of proteins in vivo.

Fig. 1: There is no information about the concentration of ST-2 (only in method section). In the method section it is claimed that various time points are analyzed. What is the time-point shown here? Does each dot indicate the mean of three replicates? Please rearrange this figure. Show those analyzed by WB separately and those analyzed by ELISA separately. Otherwise the selection is unclear. The ELISA should read ng/ml or something else not Protein (AU). This allows us to judge about the sensitivity of the ELISA. Please show more than one band per condition and a greater part of the blot including molecular weight markers. Please add a conventional loading control.

Fig. 1 E and F should be a different Figure. Here you do not control the experimental condition (as under A-D) but precisely analyze a candidate. This should be separated. WB and ELISA comments see above. Why are the n numbers so different? Did the WB or ELISA not work properly? From the design of the experiment it should not be different.

Fig. 2: It is essential to show the down-regulation of Nrp-1. 80% is not a knockout.

Fig. 3: You show different concentrations but again, what is the time point? Fig. 1A: unclear which of the samples are p<0.05 vs. control.

Fig. 4: Now we have time points but no concentration! NFkB makes me confuse. Is the upper band a different antibody? This has not be stated. Should phosphorylation not increase the MW? Is IFkB not the (phosphorylated) level of regulation and NFkB (once released) translocates to the nucleus? Is the whole cell protein? As the point is critical: Please clarify. Similarly: p42/44: What is the difference?. For clarification replace “activation” by “phosphorylation”. You did not measure activity.

Fig. 5B: Please use different symbols for those samples from AOB and sham. In the method section it was stated n=7 for both groups. Now we have n=6-13 (all at all20) but only 12 dots? Please clarify. Give r values and p values for this type on analysis and mention the test in the statistic chapter.

Discussion:

Start with a sentence like: The new finding of this study, because large parts are (necessary) controls. Why did you ignore other TGF-ß modulators such as biglycan and decorin?

Reviewer 2 Report

The manuscript by Matilla et al.is of marginal importance for understanding of the activation of cardiac fibroblasts and type I collagen synthesis in cardiac fibrosis, because it has been done without required scientific rigor and has multiple major flaws which are listed below.

  1. The concentrations of sST2 used in this work are not physiological. The plasma concentration of sST2 was measured in cardiac patients in the manuscript: “Increased Plasma Concentrations of Soluble ST2 are Predictive for 1-Year Mortality in Patients with Acute Destabilized Heart Failure” Thomas Mueller, Benjamin Dieplinger, Alfons Gegenhuber, Werner Poelz, Richard Pacher, Meinhard Haltmayer. Clinical Chemistry, Volume 54, Issue 4, 1 April 2008, Pages 752–756. It was 870 ng/L in non-surviving patients and 342 ng/L in surviving patients. This study used 2μg/ml. How can results obtained >2000-fold higher concentration be physiologically relevant? What is the significance of this work?
  2. The authors knocked down NRP in cardiac fibroblasts using CRISPR/Cas9 but the result of the knock down is not shown. How can the most important control experiment be listed as not shown, just saying that the knock down was 80%? How was this 80% estimated? Without showing the western blot of NRP in control and knock down cells this manuscript can not be published. How can you base your conclusions on a result that you did not show to the readers?
  3. Western blots throughout the manuscript are shown without loading controls. The authors stated that stain free detection was used as loading control, whatever that means. This is insufficient, especially because the changes in expression are generally less than two fold. Re-probing of the blots for actin or tubulin is essential for interpretation of the results. Also, the bands are tightly cropped, so we don’t know what else had been detected on these blots and even if these bands represent the proteins claimed, as the actual size marker is not shown. Without presenting these results with accepted scientific rigor, the manuscript should not be published.
  4. In Fig 5 it is not clear what was measured, sST2 or ST2L or both. The figure title says sST2 and the axis are labeled as ST2. If it is total ST2, how does that relate to sST2 and how does this result corroborate the other findings in the manuscript?